# Therapeutic and Reconstructive Management Options in Scleroderma (Morphea) en Coup de Sabre in Children and Adults. A Systematic Literature Review

**DOI:** 10.3390/jcm10194517

**Published:** 2021-09-29

**Authors:** Ewelina Ulc, Lidia Rudnicka, Anna Waśkiel-Burnat, Olga Warszawik-Hendzel, Anna Niemczyk, Małgorzata Olszewska

**Affiliations:** Department of Dermatology, Medical University of Warsaw, 02-008 Warsaw, Poland; ewelina.ulc@gmail.com (E.U.); lidiarudnicka@gmail.com (L.R.); waskiel.a@gmail.com (A.W.-B.); olga.warszawik@wp.pl (O.W.-H.); malgorzataolszewska@yahoo.com (M.O.)

**Keywords:** cyclosporine, hydroxychloroquine, methotrexate, morphea, scleroderma, tocilizumab, treatment, therapy

## Abstract

Scleroderma (morphea) en coup de sabre is a localized subtype restricted to the frontoparietal region of the head. Current treatment paradigms rely on low levels of evidence, primarily case reports and case series-supported by expert opinions. The aim of this article was to systematically analyze current data related to the treatment of localized scleroderma en coup de sabre. The databases Scopus, PubMed, and EBSCO were searched for all reports discussing the treatment of localized scleroderma en coup de sabre. The keywords en coup de sabre, “facial linear scleroderma”, and “morphea linearis”, combined with “treatment” or “therapy” were used as search terms. A total of 34 articles analyzed treatment outcomes for patients with localized scleroderma en coup de sabre including 4 retrospective cohort studies, 2 prospective cohort studies, 4 case series, and 24 case reports, representing a total of 69 patients (38 children and 31 adults). Methotrexate was the most commonly investigated treatment (26 patients) with a highest response rate (26/26, 100%). Other treatments included systemic glucocorticosteroids (nine patients), followed by UVA1 (four patients), mycophenolate mofetil (two patients), hydroxychloroquine (five patients), abatacept (two patients), tocilizumab (three patients), cyclosporine (one patient), interferon gamma (one patient), PUVA therapy (two patients), NB-UVB therapy (one patient), and pulsed dye laser (one patient). Reconstructive and surgery treatment was successfully used for lesions with settled disease activity to improve the cosmetic aspect of the lesions. Conclusion: methotrexate is the most often-studied treatment and reported good clinical outcomes in children and adults with localized scleroderma en coup de sabre.

## 1. Introduction

Localized scleroderma/morphea en coup de sabre (LScs) is a rare form of localized scleroderma that typically affects predominantly children and women [1]. It manifests by presence of linear atrophy and/or hardening of the skin, subcutis, occasionally involving muscles and bones [2]. The early phase lesions appear as an erythematous or violaceous linear indurated mild atrophic plaque and subsequently lesions progress to hypopigmented or depigmented sclerotic deep furrow [3]. It usually starts at the level of the upper eyebrow ridge and reaches the scalp, where a cicatricial alopecia focus appears [4]. There are known descriptions of patients with localized scleroderma en coup de sabre, in whom lesions spread below the eyebrows involving the eyelids, eyelashes, or the skin on the nose [5]. The disease may manifest with ophthalmologic (deformation of eyelids, uveitis, episcleritis) and neurological (convulsions, migraine, trigeminal neuralgia, vascular malformations) symptoms [6,7]. In some cases neurological symptoms preceded the appearance of skin lesions [8]. Parry Romberg syndrome (also known as progressive facial hemiatrophy), which is a distinct entity within craniofacial linear subtype involving subcutaneous tissue and bones, coexists in 20–40% of patients with en coup de sabre lesions [6].

The aim of the review was to systematically analyze data about the treatment efficacy of localized scleroderma en coup de sabre.

## 2. Methods

Scopus, PubMed, and EBSCO databases were searched for all reports discussing the treatment of localized scleroderma en coup de sabre published up to 4 January 2021. The keywords “en coup de sabre”, “facial linear scleroderma”, and “morphea linearis” combined with “treatment” or “therapy” or “management” were used as search terms.

Search results were analyzed and case reports, case series, and clinical trials published in the English language covering the subject of treatment of localized scleroderma en coup de sabre were included, the first one dating from 2003. Case series were defined as reports on a treatment outcome in more than two patients with localized scleroderma en coup de sabre. For each included study, reported variables such as the author, year, the type of study, the number of patients, the type of treatment, response outcomes, the frequency of relapses and side effects were recorded. Response to therapy was defined as absence of extension of the lesions and improvement at least one of the following: signs of inflammations, softening and/or lightening on the skin by clinical examination. No response to therapy was defined as largening or worsening of pre-existing lesions or new lesions. Studies were excluded if they did not include the results of treatment, had form of conference abstracts, or were a review of earlier literature data. The quality of all studies was rated based on the Oxford Centre for Evidence-Based Medicine Levels of Evidence rating scheme [9].

This article was based on previously conducted studies and does not contain any studies with human participants or animals performed by any of the authors.

## 3. Results

A total of 1109 potentially relevant unique citations were identified from our literature search (Figure 1). Of these, 320 articles were selected for further evaluation based on the relevance of their title and abstract. A total of 34 articles examining the treatment of localized scleroderma en coup de sabre were ultimately included in this study.

A total of 34 articles examined treatment outcomes for patients with localized scleroderma en coup de sabre, including 4 retrospective cohort study, 2 prospective cohort study, 4 case series, and 24 case reports representing a total of 69 patients. Three patients had extracutaneous (neurological and ophthalmic) manifestations.

A summary of detailed results is presented in Table 1 and Table 2.

### 3.1. Pharmacological Treatments

#### 3.1.1. Methotrexate

Methotrexate, a folic acid antagonist, is an immunosuppressive agent which inhibits cytokines that play an important role in sclerotic skin disease such as interleukin 2, 4 and 6 [2,32].

Systemic methotrexate was the most commonly reported monotherapy and combined with glucocorticosteroids for localized scleroderma en coup de sabre, achieving a favorable responding rate of 100% [2,10,30,31,32,33].

Rattanakaemakorn et al. conducted a retrospective study on the efficacy of methotrexate in seven patients with localized scleroderma en coup de sabre (six children and one adult). The starting dose of methotrexate was 2.5 mg weekly in pediatric patients and 10 mg weekly in the adult patient. All seven patients improved with methotrexate monotherapy. No adverse events were observed, except in one patient (14%), who developed nausea. Patients were followed for an average of 24 months [2].

Hardy et al. performed study including 12 children (<18 years) with active localized scleroderma en coup de sabre treated with methotrexate for a minimum 4 months. Methotrexate dosage ranged from 7.1 to 15 mg/m^2^/week. At evaluation, performed at a median time of 11 months after methotrexate initiation, four patients improved and eight patients remained stable [30].

Polcari et al. reported four pediatric patients with localized scleroderma en coup de sabre, who were treated methotrexate (15–25 mg/week) combined with glucocorticosteroids (prednisone, prednisolone, or methyloprednisolone). Therapy slowed disease progression in all patients, but in one of them the worsening of skin lesions after tapering of medications was observed [31].

Anderson et al. reported the case of a 10-year-old girl with localized scleroderma en coup de sabre who was treated with methylprednisolone 1000 mg weekly for 12 weeks with transition to a prednisone taper and 25 mg weekly subcutaneous methotrexate with transition to oral therapy. Her skin lesion became smaller and softer with regression of the violaceous border. Methotrexate was discontinued after nearly 3 years [32].

Niklander et al. described a case of a 13-year-old girl with localized scleroderma en coup de sabre. A treatment of prednisone 15 mg/day, methotrexate (MTX) 20 mg/week (intramuscular), and folic acid 5 mg daily after the dose of methotrexate was administered. The patient reported no adverse side effects. The drug therapy described above was maintained for 15 months. No other lesions appeared during this period and the existing ones underwent no change or became smaller [10].

Van der Veken et al. reported a case of a 19-year-old woman with mixed scleroderma (en coup de sabre and circumscribed scleroderma) who was treated with methotrexate. An initial dose of 7.5 mg/week was well tolerated and therefore increased to 15 mg/week after 2 weeks. This treatment was continued for 12 months. The goal of the treatment was to stop progression of the skin lesions. The only adverse effects were a subjective feeling of fatigue and abdominal discomfort.

#### 3.1.2. Systemic Glucocorticosteroids

Glucocorticosteroids have anti-inflammatory and immunosuppressive effects and also anti-proliferative effects on keratinocytes. Furthermore, they can suppress collagen synthesis by fibroblasts [43].

A study performed by Joly et al. revealed that systemic glucocorticosteroids were effective in 100% (7/7) patients with localized scleroderma en coup de sabre. Patients were given doses of 0.5–1 mg/kg prednisone per day for 6 weeks, followed by a progressive decrease of the dosage. Treatment lasted 5 to 70 months (mean, 18.3 months); four patients improved and three patients remained stable. The follow up period lasted at least 18 months after the initiation of therapy. There were no data how many patients with localized scleroderma en coup de sabre had recurrence of skin lesions after treatment discontinuation [11].

Arif et al. described a 17-year-old girl with concomitant localized scleroderma en coup de sabre and plaque type scleroderma. Treatment for the patient was prednisolone 30 mg/day and topical tacrolimus ointment 0.1% to be applied twice daily. After 1 month of treatment, there were no new lesions and the existing lesions over the scalp and thigh showed improvement with respect to thickness and pigmentation [34].

Unterberger et al. reported a 24-year-old woman in the 33rd week of pregnancy who developed right-sided hemiparesis and progressive neurological complications in association with localized scleroderma en coup de sabre. The patient was treated with a pulsed intravenous methylprednisolone, 1 g daily for three days, followed by 500 mg daily for a further three days, and a consecutive tapering period with oral methylprednisolone. During the following two weeks this treatment led to a rapid improvement in the neurological symptoms. The patient recovered well and was discharged five weeks after the onset of her neurological symptoms [12].

#### 3.1.3. Cyclosporine

Cyclosporine (a calcineurin inhibitor) selectively inhibits the release of IL-2 from activated lymphocytes [44].

There is one case report of a 7-year-old girl with localized scleroderma en coup de sabre, who was treated with oral cyclosporine 3 mg/kg/day for 3 months. Skin lesions improved 3 months after starting treatment but relapsed 18 months after she completed treatment. Adverse drug reactions were not reported [35].

#### 3.1.4. Mycophenolate Mofetil

Mycophenolate mofetil exerts an inhibitory effect on T- and B-lymphocyte proliferation. It showed expression of inhibit type I collagen to enhance the expression of matrix metalloproteinase-1 and to alter both the migratory and contractile functions of fibroblasts. Thus, mycophenolate mofetil has direct antifibrotic properties in addition to its well-known immunosuppressive effects.

Martini et al. conducted a retrospective study on the efficacy of mycophenolate mofetil in 10 pediatric patients with localized scleroderma, of which 2 had en coup de sabre lesions. In one patient (3-year-old boy) with en coup de sabre lesions, mycophenolate mofetil was chosen as the first therapy because besides skin lesion activation, concomitant cerebral and ocular vasculitis was present. In second patient (6.5-year-old boy) the first treatment was intravenous methylprednisolone, oral prednisone, and methotrexate (15 mg/m^2^). The mean duration of treatment with mycophenolate mofetil, at the last follow-up evaluation, was 20 months. The daily dose was 600–1200 mg/m^2^/day, twice daily. In both patients, arrest of disease progression and reduction of erythema were observed. The patient with scleroderma en coup de sabre associated with ocular and cerebral vasculitis exhibited markedly improved ophthalmological examination and an arrest of progression of cerebral vasculitis as shown by MRI (Magnetic Resonance Imaging) [13].

#### 3.1.5. Hydroxychloroquine

Hydroxychloroquine has antithrombotic and antifibrotic properties [45,46]. Kumar et al. investigated the effect of hydroxychloroquine in a group of five patients with localized scleroderma en coup de sabre (four of them were younger than 18 years). The daily dose for adult patients was 400 mg and 5 mg/kg for children. Treatment with hydroxychloroquine monotherapy lasted for a minimum of 6 months. Clinical outcomes of treatment were classified as follows: complete response indicated total resolution of active skin lesions and lack of new lesions, or at least 95% improvement as qualitatively graded by the physician; partial response indicated persistence of some active skin lesions (with or without the development of new lesions), with resolution of some lesions such that extent or severity was decreased (partial response was graded as >50% or ≤50%); no response indicated persistence, worsening, or increase in skin lesions; and relapse was defined as the appearance of skin lesions at the same sites or at different sites 1 year or more after complete response. Two patients with localized scleroderma en coup de sabre had a partial response greater than 50%, two had partial response less than or equal to 50%, and one had no response [17]. No side effects or relapses were reported in patients with localized scleroderma en coup de sabre treated with hydroxychloroquine.

#### 3.1.6. Abatacept

Abatacept is a soluble recombinant fusion protein that inhibits T-cell activation by binding to CD80 and CD86, thereby blocking interaction with CD28 [47].

Fage et al. presented data concerning two patients with localized scleroderma en coup de sabre treated with abatacept intravenously (500–750 mg/day on days 1, 15, 30, and thereafter every 4–6 weeks) for 3–21 months.

A significant reduction of lesions, approximately 50%, of the area (measured in cm^2^) was observed in both patients. Adverse events associated with abatacept use were reported in 1 patient and included oral aphthous ulcers [14].

#### 3.1.7. Tocilizumab

Tocilizumab is a monoclonal antibody raised against the soluble receptor for interleukin 6. IL-6 inhibition can be an effective target in localized scleroderma, as serum levels of soluble IL-6 receptor have been found to be increased in patients with localized scleroderma compared with healthy controls [48,49]. In two published case reports and one case series, which included three patients with en coup de sabre, subcutaneous (162 mg/weekly in one patient) [16] or intravenously (8–10 mg/kg every 3–4 weeks in two patients) [36,37], tocilizumab was effective in all of them. Adverse drug reactions and relapse rate were not reported.

#### 3.1.8. Interferon Gamma

Interferon gamma (IFN-ϒ) is a dimerized soluble cytokine, which has a strong inhibitory effect on collagen synthesis by normal dermal and scleroderma fibroblasts in vitro. In addition, inhibition of growth and chemotaxis fibroblasts, and decrease of fibroblasts adhesion to collagens I, IV, VI, fibronectin, and laminin has been described [50]. One case report described a patient with localized scleroderma en coup de sabre complicated by orbital involvement who was successfully treated with interferon gamma (100 mg 3 times a week subcutaneously—52 mg/m^2^ body surface area). Adverse drug reactions and relapse rate were not reported [15].

#### 3.1.9. UVA1-Therapy

The specific mechanism of action of ultraviolet therapy in the treatment of localized scleroderma is unknown. Studies indicate that UVA1 causes apoptosis of epidermal Langerhans cells and T cells. UVA1 also affects fibroblasts, increasing synthesis of collagenases and decreasing synthesis of collagen. It is also thought to impair collagen cross linking. UVA1 also affects levels of local cytokines. It causes a decrease in interleukin-6, which decreases collagen and glycosaminoglycans, a decrease in transforming growth factor beta (TGFb), which decreases fibroblast growth, and an increase in IFN-ϒ, which increases matrix metalloproteinase-1 [51].

Su et al. described three patients with localized scleroderma en coup de sabre in which UVA1-therapy (30 J/cm^2^ 3–5 times a week for 10–15 weeks) was effective. In two patients, relapse was observed after stopping the therapy [18]. Kowalzick at al. reported the case of a 13-year-old girl who was treated with a combination of topical calcipotriol 0.005% twice a day and 30 daily whole-body irradiations each of 30 J/cm^2^ UVA-1. After that treatment, a stop of the progression and a softening of the lesion was achieved [19]. Side effects were not reported.

#### 3.1.10. PUVA-Therapy

PUVA may suppress collagen synthesis and induce collagenase activity, resulting in clinically observed softening of former sclerotic lesions. Gambichler et al. described two patients with localized scleroderma en coup de sabre treated with topical calcipotriol and cream psoralen plus ultraviolet A. The initial UVA dose was 0.3 J/cm^2^. The treatment was performed three times weekly and the UVA dose was increased at the earliest after 3 days with 0.2 J/cm^2^. In addition, calcipotriol ointment was applied twice daily. Three months after beginning therapy, a considerable softening of sclerotic lesions was observed in both patients. Forty treatments resulted in a cumulative UVA dose of 71 J/cm^2^. No side effects were observed [38].

#### 3.1.11. NB-UVB Therapy

Narrow-band (NB-UVB) phototherapy is used to treat inflammatory and T-cell mediated dermatoses. Browned et al. investigated the effect of narrow-band ultraviolet B (NB-UVB) phototherapy combined with antimalarials in one 32-year-old female with localized scleroderma en coup de sabre. Therapy slowed disease progression and reversed hair loss. One year after stopping NB-UVB therapy, the progression of her condition with the onset of alopecia of the affected scalp was noted. No adverse effects were observed [20].

#### 3.1.12. Pulsed Dye Laser (595 nm)

Pulsed dye laser is the gold standard for treatment of port wine stains. These lasers selectively target hemoglobin, resulting in destruction of dilated ectatic capillaries in the upper dermis, while sparing the surrounding tissue [52]. Kakimoto et al. reported the case of a 6-year-old girl with localized scleroderma en coup de sabre who was given an initial diagnosis of acquired port wine stain and treated with a pulsed dye laser (595 nm) [39]. Pulsed dye laser therapy combined with topical clobetasol helped to alleviate initial erythema. Two years later, a relapse of skin lesions was noted. The side effects included blistering and hypopigmentation.

### 3.2. Reconstructive Treatments

#### 3.2.1. Fat Grafting

Reconstructive treatment is proposed as an option for lesions with settled disease activity to improve the cosmetic aspect of lesions [53]. Structural fat grafting is ideal for the correction of localized tissue atrophy or craniofacial deformities. Adipose tissue contains the highest percentage of stem cells, and therefore structural fat grafting creates new vascularization and real structural alterations [24]. According to the present review, fat grafting were effective in all patients (6/6). No side effects or relapses were reported in patients with localized scleroderma en coup de sabre treated with fat grafting [21,22,23,25,40].

#### 3.2.2. Hyaluronic Acid Filler

Hyaluronic acid filler is particularly well suited for soft tissue augmentation because of its tolerability, availability, relatively low cost, reversibility, and efficacy in volumization. In addition to these benefits, it plays a role in cell growth, membrane receptor function, and adhesion [54]. Upon review of the literature, we encountered two cases in which hyaluronic acid filler was utilized in the correction of patients with localized scleroderma en coup de sabre with successful improvements [26,27].

#### 3.2.3. Polymethylmethacrylate

Polymethylmethacrylate is a permanent filler, biocompatible, non-toxic, non-mutagenic, and immunologically inert. After polymethylmethacrylate application, there is a reaction of a foreign body type that induces the onset of giant cells that wrap each particle of the product, leading to new collagen and blood vessels formation [55]. Franco et al. reported the case of a 14-year-old male patient with localized scleroderma en coup de sabre who was treated with polymethylmethacrylate. After the treatment, they observed raising of the depressed portion and partial hair regrowth in the alopecia area of the scalp [41].

#### 3.2.4. Tissue Cocktail Injection

Tissue cocktail is a minced amalgam of dermis, fascia, and fat obtained from de-epithelialized soft-tissue components. Oh et al. presented a case of trilinear scleroderma en coup de sabre that was corrected by autologous tissue cocktail injection with excellent cosmetic results. No complication was observed at the 18 month follow-up, and excellent cosmetic results were achieved [56].

#### 3.2.5. Toxinum Botulinum

Botulinum toxin inhibits the release of neuropeptides, such as substance P and glutamate, which are involved in the regulation of pain and inflammation [28]. One possible mechanism of the skin lesions of localized scleroderma en coup de sabre is neural-based vasoconstriction causing atrophy of skin, muscle, periosteum, and bone. By blocking the transmission of norepinephrine, botulinum toxin prevents the signal that causes vasoconstriction in vascular smooth muscle. Rimoin at al. reported the efficacy of botulinum toxin in improving disfigurement and headaches associated with scleroderma en coup de sabre [29].

#### 3.2.6. Surgical Treatment

Surgical removal of the band-shaped sclerosis is possible as an alternative therapy of localized scleroderma en coup de sabre. There was one case of a patient with localized scleroderma en coup de sabre who was successfully treated with an en bloc resection of the band-shaped sclerotic area [42].

## 4. Discussion

The evaluation of treatment efficacy in localized scleroderma en coup de sabre is difficult considering the fact that it based only on studies with moderate and low levels of evidence (III-V). They may have both many bias and confounders. The high risk of bias in observational studies results from a number of factors, particularly the lack of control treatment or unclear design and assessor bias, especially in small cohorts of patients. Moreover, data that describe no effects of treatment may be unpublished. 

Of all the treatment modalities described in this study, methotrexate and glucocorticosteroids were used to treat the greatest number of patients with the response rate of 100%. The assessment of the efficiency of UVA1 therapy was based on studies with a moderate level of efficacy (III) and characterized by response rates of 100% and high relapse rates after the end of the treatment (67%). Other commonly suggested therapies including mycophenolate mofetil, hydroxychloroquine, abatacept, tocilizumab, cyclosporine, and interferon gamma were used in cases of contraindications methotrexate or intolerance and affected a small number of patients with localized scleroderma en coup de sabre. PUVA therapy, NB-UVB therapy, and pulsed dye laser were effective in some cases. Reconstructive and surgery treatment were successfully used for lesions with settled disease activity to improve the cosmetic aspect of the lesions. 

European Dermatology Forum experts recommend methotrexate in monotherapy or in combination with systemic glucocorticosteroids (methyloprednisolone or prednisone) in the treatment of localized scleroderma of en coup de sabre [57]. Methotrexate was extensively studied in numerous studies and its efficacy was confirmed in randomized controlled trials [58]. Mycophenolate mofetil should be considered as a second-line therapy if methotrexate fails. Other therapies including cyclosporine A, azathioprine, chloroquine and hydroxychloroquine, intravenous immunoglobulins, abatacept, infliximab, rituximab, or imatinib should be reserved to single, severe cases with contraindications or failure of standard therapy [57]. 

Plastic-surgical intervention in en coup de sabre lesion is an option to improve cosmetic aspects of the lesions. In order to minimize the risk for flare-ups, it is imperative that surgery only be performed in the inactive stage of the disease, ideally several years after the end of disease activity [59].

The choice of therapeutic options for patients with localized scleroderma en coup de sabre should be preceded by examination of the patient and assessment of the disease activity and intensity. Immunosuppressive therapies are recommended as early as possible and long enough to reach a non-active phase [8].

Ideally, a randomized clinical trial with long-term follow-up and objective measures should be performed to investigate the therapies presented in this study and to describe the outcomes.

## 5. Conclusions

There is a paucity of high-quality studies on which to base treatment recommendations for localized scleroderma en coup de sabre. Although the evidence is limited, the results of this review suggested that methotrexate in monotherapy or combined with systemic glucocorticosteroids (added in the induction phase) were the most well-studied treatments and reported good clinical outcomes both in children and adults.

Other treatment options such mycophenolate mofetil, cyclosporine, and biologics therapies (abatacept, tocilizumab, interferon gamma) were effective in cases of refractory localized scleroderma en coup de sabre. Reconstructive surgery and hyaluronic acid fillers were used to improve the cosmetic aspect of the lesions. The results of this systematic review may provide a guideline for the clinical practice and may be used to guide future multicenter studies to verify the efficacy of these treatments modalities.

## Figures and Tables

**Figure 1 jcm-10-04517-f001:**
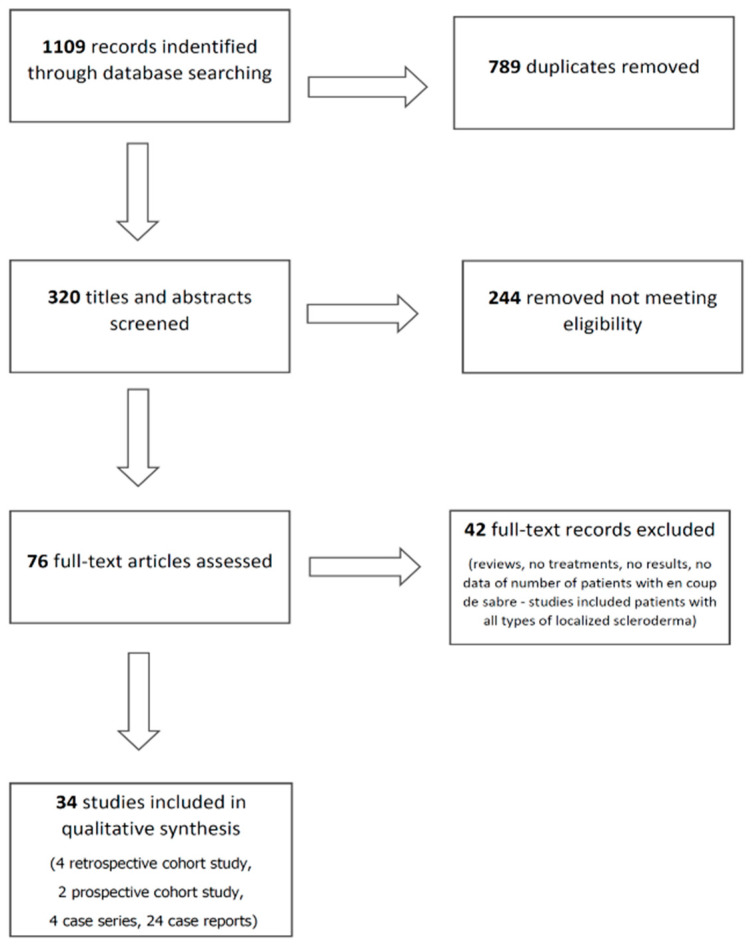
PRISMA Flowchart.

**Table 1 jcm-10-04517-t001:** Summary of studies regarding treatment of localized scleroderma en coup de sabre in adults.

Medicament	Dose	Number of Patients with Localized Scleroderma en Coup de Sabre	Response Rate	Highest Level of Evidence	Literature
Methotrexate	15 mg/week	1	100% (1/1)	IV	Rattanakaemakorn P, et al. [2]
Methotrexate	15 mg/week	1	100% (1/1)	V	Van der Veken D, et al. [10]
Prednisone	0.5–1 mg/kg/day	7	100% (7/7)	III	Joly P, et al. [11]
Methylprednisolone	1 g/day for three days followed by 500 mg/day for further three days intravenosus and tapering with oral methylprednisolone	1	100% (1/1)	V	Unterberger I, et al. [12]
Mycophenolate mofetil	600–1200 mg/m^2^/day	2	100% (2/2)	IV	Martini G, et al. [13]
Abatacept	500 mg (patients weighing <60 kg) or 750 mg (>60 kg) intravenously on days 1, 15, 30 and thereafter every 4–6 weeks	2	100% (2/2)	IV	Fage S, et al. [14]
Interferon gamma	100 mg three times a week subcutaneous (52 mg/m^2^ body surface area)	1	100% (1/1)	V	Obermoser G, et al. [15]
Tocilizumab	162 mg/week	1	100% (1/1)	V	Margo CM, et al. [16]
Hydroxychloroquine	400 mg/day	1	No data (total response 80%, 4/5, children + adults)	IV	Kumar AB, et al. [17]
UVA1 therapy	30 J/cm^2^	3	100% (3/3)	III	Su O, et al. [18]
UVA1 therapy	30 J/cm^2^	1	100% (1/1)	V	Kowalzick L, et al. [19]
NB-UVB therapy	Three times weekly	1	100% (1/1)	V	Brownell I, et al. [20]
Dermal fat grafting	No data	1	100% (1/1)	V	Barin EZ, et al. [21]
Regenerative cell-enriched autologous fat grafting	No data	1	100% (1/1)	V	Karaaltin MV, et al. [22]
Medpor with dermal fat grafting	No data	1	100% (1/1)	V	Kim KT, et al. [23]
Structural fat grafting	No data	1	100% (1/1)	V	Consorti G, et al. [24]
Alloplastic implantation with AlloDerm tissue matrix	No data	1	100% (1/1)	V	Robitschek J, et al. [25]
Hyaluronic acid	No data	1	100% (1/1)	V	Thareja SK, et al. [26]
Hyaluronic acid injected with a blunt-tipped microcannula	No data	1	100% (1/1)	V	Sivek R, et al. [27]
Tissue cocktail injection	No data	1	100% (1/1)	V	Oh HM, et al. [28]
Toxinum botulinum	25 units and 35 units at 6 and 9 months	1	100% (1/1)	V	Rimoin L, et al. [29]

**Table 2 jcm-10-04517-t002:** Summary of studies regarding treatment of localized scleroderma en coup de sabre in children.

Medicament	Dose	Number of Patients with Scleroderma en Coup de Sabre	Response Rate	Highest Level of Evidence	Literature
Methotrexate	2.5 mg/week	6	100% (6/6)	IV	Rattanakaemakorn P, et al. [2]
Methotrexate	7.1–15 mg/m^2^/week	12	100% (12/12) 8/12 (67%) stability 4/12 (33%) improvement	IV	Hardy J, et al. [30]
Methotrexate	25 mg/week (2/4) 20 mg/week (1/4) 15 mg/week (1/4)	4	100% (4/4)	IV	Polcari I, et al. [31]
Methotrexate	25 mg/week	1	100% (1/1)	V	Anderson LE, et al. [32]
Methotrexate	20 mg/week	1	100% (1/1)	V	Niklander S, et al. [33]
Prednisolone	30 mg/day	1	100% (1/1)	V	Arif T, et al. [34]
Cyclosporine	3 mg/kg/day for 3 months and 2.5 mg/kg/day for next 4 months	1	100% (1/1)	V	Crespo MP, et al. [35]
Tocilizumab	8–10 mg/kg every 3–4 weeks	1	100% (1/1)	IV	Foeldvari I, et al. [36]
Tocilizumab	10 mg\in 4 weeks	1	100% (1/1)	V	Osminina M, et al. [37]
Hydroxychloroquine	5 mg/kg/day	4	general response 80%, (4/5, children + adults)	IV	Kumar AB, et al. [17]
PUVA therapy	Topical 8-methoxypsoralen 0.0006%; The initial UVA dose was 0.3 J/cm^2^ three times weekly (increased by 0.2 j/cm^2^ after 3 days)	2	100% (2/2)	IV	Gambichler T, et al. [38]
Pulsed dye laser (595 nm)	8-J/cm^2^ fluence, 1.5 ms pulse duration, 40 ms duration, and 30 ms delay with a dynamic cooling, the fluence was increased to 8.5 J/cm^2^ and the dynamic cooling increased to 50/30	1	100% (1/1)	V	Kakimoto CV, et al. [39]
Autologous fat grafting	No data	1	100% (1/1)	V	Ayoub R, et al. [40]
Polymethylmethacrylate	No data	1	100% (1/1)	V	Franco JP, et al. [41]
Resection of the sclerotic area	No data	1	100% (1/1)	V	Dirschka T, et al. [42]

## Data Availability

The data presented in this study are available on request from the corresponding author.

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
