# Peer review of "Therapeutic and Reconstructive Management Options in Scleroderma (Morphea) en Coup de Sabre in Children and Adults. A Systematic Literature Review"

_jcm, 2021, doi:10.3390/jcm10194517_

Round 1
Reviewer 1 Report
Dear Authors, Thank you for the possibility to review your fine paper. You have done a thorough work and the paper is indeed of interest to all of us who diagnose and treat these patients. I agree that the primary conclusion may be that it calls for multicenter randomized controlled trials involving several centers in plural countries.
I have a few minor comments that may be of more help:
- Your heading: Reading through your paper it becomes obvious that you review treatment efficay in a broader sence: Both treatments to provide remission and treatments to improve cosmetic outcomes. Could you improve your heading so it is obvious to the reader?
- Method - how old is the oldest papers included in the review?
- Discussion - fine first lines but I suggest that you add just a little concerning specifically publication bias. Case series and retrospective studies may have both many bias and confounders. The greatest problem of the all may however be that papers/data, that describes no effect is unpublished. Just one or two sentences concerning this problem will improve the your discussion even further.
- I notice that you use the 2017 European Dermatology Forum expert guideline as a reference concerning the statement: 'Plastic Surgery ... are only considered in inactive stage of the disease to keep down reactivation of the disease'. Do we really know that plastic surgery does keep down reaction? Clinically I have experienced otherwise and I suggest that you find another reference to support this statement , if you keep it in the review. Good luck, all the best
Reviewer 2 Report
General comments:
I feel this is best described as a narrative review or data synthesis rather than a systematic review.
Please ensure consistency of terminology throughout the manuscript.
The condition being studied is Localized Scleroderma (also known as morphea), linear subtype with craniofacial involvement, specifically en coup de sabre lesion.
Specific comments:
Intro:
Line 42: Parry Romberg syndrome (also known as hemifacial atrophy) is a distinct entity within craniofacial linear subtype, involving deeper structures.
Methods:
Line 57: Response to therapy was defined as decrease in size of the initial lesions and/or 57 skin softening without new. This is not a complete sentence.
Results:
It might be worth indicating somewhere (in text or table) if/how many patients had extra-cutaneous manifestations or not.
Line 83-4: It seems unlikely that all reported studies with MTX treatment report a 100% response rate.
Line 91: There is no mention of total sample size in Hardy et al study making it hard to interpret the reported results.
Table 1: there is inconsistency between the number of patients listed in the table and the number written in the corresponding text.
Line 201: Clarify if tocilizumab was effective in all 3 patients
Line 221: For how long did the patients have UVA therapy?
Suggest separating the medical treatments for active lesions and the reconstructive treatments for damage/ tissue loss.
Discussion:
May want to mention the one RCT for MTX in localized scleroderma Zulian F, et al. Methotrexate treatment in juvenile localized scleroderma: a randomized, double-blind, placebo-controlled trial. Arthritis Rheum. 2011 Jul;63(7):1998-2006. Unfortunately it does not distinguish ECS patients.
